# From Leaves to Reproductive Organs: Chemodiversity and Chemophenetics of Essential Oils as Important Tools to Evaluate *Piper mollicomum* Kunth Chemical Ecology Relevance in the Neotropics

**DOI:** 10.3390/plants13172497

**Published:** 2024-09-06

**Authors:** Daniel de Brito Machado, Jéssica Sales Felisberto, George Azevedo de Queiroz, Elsie Franklin Guimarães, Ygor Jessé Ramos, Davyson de Lima Moreira

**Affiliations:** 1Graduate Program in Plant Biology, Institute of Biology, State University of Rio de Janeiro, Maracanã, Rio de Janeiro 20550-013, Brazil; jessickasalles@gmail.com (J.S.F.); ygorjesse@gmail.com (Y.J.R.); 2Rio de Janeiro Botanical Garden Research Institute, Botanical Garden of Rio de Janeiro, Rio de Janeiro 22460-030, Brazil; eguimar@jbrj.gov.br; 3West Zone Campus, State University of Rio de Janeiro, Rua Manuel Caldeira de Alvarenga, Rio de Janeiro 23070-200, Brazil; georgeazevedo08@gmail.com; 4Earth’s Pharmacy Laboratory, School of Pharmacy, Federal University of Bahia, Salvador 40170-215, Brazil; 5Oswaldo Cruz Foundation, Farmanguinhos, Manguinhos, Rio de Janeiro 21041-250, Brazil

**Keywords:** Piperaceae, Atlantic Forest, essential oils, monoterpenes, sesquiterpenes

## Abstract

*Piper mollicomum* Kunth (Piperaceae) plays a vital role in the preservation of the Brazilian Atlantic Forest by contributing to the regeneration of deforested areas. Recent scientific investigations have analyzed the chemical constituents and seasonal dynamics of essential oils (EO) from various *Piper* L. species, highlighting the need to elucidate their chemical–ecological interactions. This study aims to expand the chemical–ecological knowledge of this important taxon in neotropical forests, using *P. mollicomum* as a model. The methodologies employed include the collection of plant material, EO extraction by hydrodistillation, analysis of EO by gas chromatography–mass spectrometry (GC–MS) and gas chromatography–flame ionization detector (GC–FID), recording the frequency of visits by potential pollinators and microclimatic variables, and by conducting calculations of chemodiversity and chemophenetic indices. Chemical analyses indicated that the diversity of EO and environmental factors are linked to the activities of potential pollinators. In the Tijuca Forest, *P. mollicomum* revealed significant interactions between its volatile constituents and microclimatic variables, showing that the chemodiversity of the leaves and reproductive organs correlates with pollinator visitation. Additionally, a notable difference in chemical evenness was observed between these vegetative structures. The chemophenetic indices by Ramos and Moreira also revealed correlations with chemical diversity.

## 1. Introduction

A brief recreational stroll through a garden, or even exhaustive fieldwork, can offer much more than just a visual experience, as, in these environments, our senses are stimulated in diverse ways. For example, while walking along a path adorned with woods or herbs, we may encounter anything from the fresh and inviting aroma of mint (*Mentha piperita* L.) to the unpleasant and repulsive odor of titan arum (*Amorphophallus titanium* (Becc.) Becc. ex Arcang, 1878). These aromas are not only perceived by humans; they are samples of a complex and elegant set of chemical languages used in plant–plant, plant–insect, and plant–microorganism communication, essential for the integration of plant species into their different niches [1,2,3,4,5,6,7,8,9,10,11]. Informed by these insightful understandings, can we infer that these constituents have always been tasked with the same ecological activities? If the answer is yes, what has contributed, retrospectively, to the current patterns of disparity, regularity, occurrence, and chemical diversity of these metabolites? Regardless, we know there are some approaches on the subject at hand, as will be discussed below.

Chemical diversity refers to the array of constituents present at different levels of organization in a specific ecosystem [12]. It is worth noting that tropical forests, for instance, have played a pivotal role as “living pharmacies”, offering a rich source of novel substances [12,13]. Therefore, understanding the dynamics of metabolites present in these areas is crucial for advancing the field of human health [14]. Basic research in natural products, with an emphasis on plant studies, is highly relevant because the wide range of plant constituents play important roles in the therapeutic process [15,16].

Chemodiversity is a metric used to quantify the chemical diversity of a population, and it is subject to a range of influences, from environmental conditions (abiotic) and genetic factors to biotic interactions [11,12]. For a long time, the influence of these chemical constituents on the attraction of pollinating insects was considered only in specific cases of coevolution or treated in a stochastic manner. Today it is known that these volatiles directly and indirectly influence the network of ecological interactions to which plant species are subjected [11,16,17,18,19,20,21]. Therefore, it is crucial to study taxa that are influenced by these interactions to understand trophic and other ecological interactions in their habitat. In this context, the genus *Piper* L. stands out.

The Atlantic Forest, a coastal biome stretching from Northeastern Brazil to Southeastern Argentina, stands out for its phytophysiognomic diversity, adapted to variations in climate, soil, and topography. Despite harboring rich biodiversity, with over 15,000 plant species, including the Piperaceae family, it faces severe threats from uncontrolled urbanization, fragmentation, deforestation, and pollution. With less than 10% of its original cover remaining, competition for resources and edge effects compromise its stability. Nevertheless, the Atlantic Forest plays a crucial role in climate regulation, water conservation, and disaster mitigation, making it essential for Brazil’s socio-economic sustainability [11,16].

*Piper mollicomum* Kunth is a plant species that allocates a significant portion of its energy to the biosynthesis of volatile building blocks for the establishment of their ecological interactions [11,22,23,24,25,26,27]. This species occurs in various tropical forests across Brazilian territory, ranging from high-light environments to shaded, humid locations [27,28,29,30]. In addition to their bioactive effects [31,32,33,34], these plants play vital ecological roles in ecosystems, for example, as pioneer plants in secondary forests [11,35,36]. Additionally, their inflorescences and infructescences constitute important nutritional hotspots, scarce in forests degraded by human activities, and are essential for maintaining the population density of generalist pollinators and native seed dispersers [11,35,36,37].

Ramos and collaborators published a comprehensive analysis of the chemical composition of essential oils (EO) from different organs of *P. mollicomum* from different regions of Brazil [30]. Their study showed the relevant presence of arylpropanoids in the roots, while mono- and sesquiterpenes predominated in the aerial parts of the plant. It is worth noting that this research marked the beginning of a pioneering investigation into the spatial and temporal patterns of compound biosynthesis in *Piper* L species. Additionally, aspects of the volatile chemistry of *P. mollicomum* and *Piper aduncum* L. were elaborated in a spatial-temporal scales and intra-plant context, which demonstrated substantial disparities in relation to collection sites, time (circadian cycle and seasonality), and organs of these species [30]. In addition to these two species, the study of the ecophysiology of *Piper gaudichaudianum* Kunth has been the central focus of recent research. For example, it has been recorded that phenotypic plasticity in the chemical composition of its EO, on a spatiotemporal scale, is directly associated with the level of oxidation of these volatile mixtures [26].

Continuing these research efforts, in 2022 De Brito-Machado and collaborators published the results of their investigation into the seasonal variation of the EO in leaves, inflorescences, and infructescences of *P. mollicomum* during its reproductive period. Their results provide important insights into the chemical variability among these different plant organs, but no significant correlations were found between the major volatile constituents and the attraction of potential pollinators [11].

This research, unlike previous studies, focuses on the influence of the chemical diversity of EO on the attraction of potential pollinators and how abiotic variables can modify this interaction. It introduces a scientific innovation by suggesting that, as the volatile mixture becomes richer and more diverse, it tends to become proportionally more oxidized. To expand chemical–ecological knowledge about this important taxon in neotropical forests, it is crucial to continue these investigations. Therefore, it is essential to assess the correlations between microclimatic variations and the chemodiversity of EO from different organs of *P. mollicomum*, as well as to gather more information on the influence of the chemical profile on pollinator visitation frequency.

## 2. Results

### 2.1. Chemical Composition of the Essential Oils of P. mollicomum

The main objective of this research was to investigate whether differences in the chemodiversity of the EO from the leaves and four distinct stages of the reproductive organ in *P. mollicomum* (Appendix A) are related to the attraction of potential pollinators. The results reveal variations in the yields of the EO across the four stages, ranging from 0.01% to 1.12% (*w*/*w*) (Appendix A). Additionally, variability in the chemical profile of the volatile constituents was observed during the reproductive period of the plant. Over a period of months, there was a predominance of non-oxygenated sesquiterpenes in the leaves and oxygenated monoterpenes at the different stages of the reproductive organ.

During the early months of flowering (September and October), the monoterpene linalool recorded the highest relative percentage in the leaves (44.84% and 27.20%, respectively), as well as in most stages of the reproductive organs (29.24% to 73.14%). Another notable monoterpene was limonene, which was found in a relatively high percentage (3.12% to 17.08%). In November, the predominant constituents in the leaves were identified as α-terpineol (11.41%) and β-elemene (19.95%), while 1,8-cineole showed the highest relative percentage at the different stages of the reproductive organs (26.07% to 58.77%). In December, the major constituents in the leaves were α-pinene (11.30%) and β-pinene (6.49%), as well as linalool (11.28%) and β-elemene (8.62%). On the other hand, in the reproductive organ stages, the main constituent was 1,8-cineole (22.96% to 32.01%), along with the monoterpenes α-pinene (3.50% to 14.70%) and linalool (0.00% to 10.46%). In January, the last month of the research, 1,8-cineole continued to be the constituent with the highest relative percentages in the organs, both in the leaves and at the different stages (8.62% to 49.32%). β-elemene also showed a high relative percentage content in the leaf organ (11.55%). Appendix A present the chemical constituents that, in any month of the analysis, recorded a relative percentage content greater than or equal to 5%. However, all chemicals of the EO from the leaves and different reproductive stages are included in Appendix A. It is worth noting that, for the calculation of chemodiversity and Ramos and Moreira (R&M) indices, all identified compounds were used, regardless of their relative percentage content.

### 2.2. Chemodiversity Variability of the Organs of P. mollicomum: A Multifaceted Analysis of the Leaves and Reproductive Organs

Figure 1 highlights the mean variations between the chemodiversity α indices in the mentioned leaves and reproductive organs. The graphical representation shows that the leaves exhibited significantly higher values for chemodiversity α, considering the Shannon index (*p* < 0.05), a relevant chemical richness in its EO compared with the different stages of the reproductive organs. The other chemodiversity indices did not show significant differences (*p* > 0.05) when compared between leaves and reproductive organs.

As there was a significant difference in chemodiversity between the leaves and reproductive organs, considering the Shannon index by ANOVA (Figure 1), we applied the Tukey Post-Hoc test to compare these different compartments (Appendix A and Figure 2). These analyses revealed that the chemodiversity of the EO from the leaves and stage III (anthesis) showed significant similarity (*p* = 0.103). However, there is a significant difference (*p* < 0.05) between leaves and the other stages of the reproductive organ (I, II, and IV), with the EO from the leaves being the most chemodiverse compared with these stages. Additionally, the EO from stage IV (which presents immature drupes) has the lowest richness in constituents. The average richness given by the Shannon index of constituents between stages I to IV was not statistically different (*p* > 0.05) (Appendix A and Figure 2).

Although the richness of the samples may be comparatively equal, how equal, in terms of chemical constituents, are they actually? To answer this question, the Jaccard index was used, aiming to deepen the study of chemodiversity concerning the dichotomy of the chemical profile among the samples studied. This index was used because it represents chemodiversity β and it was applied to the reproductive structures to compare the similarity between chemical constituents [38]. The results confirm the low similarity between stages I and II (53%), and between stages I and III (50%). Between stages I and IV, the similarity was even lower (36%), indicating that the development of the reproductive organ leads to chemical phenotypic diversity (Figure 3). The data from this analysis indicate chemical similarity between one stage and the next, as evidenced by the Jaccard indices between stages II and III (55%) and III and IV (55%). However, there was a gradual decrease in similarity between subsequent stages (stage II and IV—41%). This suggests that, as development progresses, the chemical similarity between stages tends to decrease.

With the aim of deepening the analyses and gaining comprehensive insights into the relationship between the gradual decrease in chemodiversity from the initial stages of inflorescence maturation to fruiting, this research proposed an innovative scatterplot to represent the interaction between richness (Shannon index) and evenness (Pielou index), as illustrated in Figure 4. This analysis confirmed that the leaves were the only organ to exhibit high levels of richness and chemical uniformity, mainly as the most prominent when compared with all other reproductive organs.

### 2.3. Chemical Ecology of P. mollicomum: Interactions between Chemical Diversity, Potential Pollinators, Volatile Compounds, and Microclimatic Variables

The data from potential pollinator observations show that bees and flies were the most frequently observed insect groups that visited *P. mollicomum* blooming inflorescences in 4332 instances (Figure 5). Bees were the most incident visitors, showing high visits on all surveyed days. The most frequent floral visitor, *Tetragonisca angustula* Latreille, 1811, was recorded as having a prominent number of visits, totaling 3042 interactions. Another Hymenoptera, belonging to the Colletidae insect family, stood out with a considerable 820 visits. Two other bee species belonging to the family Halictidae were recorded with frequencies of 238 and 67, respectively. Dipteran insects, on the other hand, showed distinct visitation rates, as one species of the Syrphidae family was the fourth most frequent insect, with 164 visits while another was observed only once on the blooming inflorescences, at the beginning of flowering (Figure 5).

Figure 6 shows the Pearson analyses carried out to assess the potential correlations between the variability in microclimate data and the frequency of visits by potential pollinators versus chemodiversity and the Ramos and Moreira indices (R&M). For instance, a strong positive correlation was identified between the R&M of stage III and the chemodiversity α indices of the leaves (*p* < 0.05). As the diversity, richness, and evenness of volatile constituents increased, the EO compounds of the flowering inflorescence at anthesis (stage III), as well as that of the leaves, tend to become more oxidized. A positive correlation between the R&M in the early stages of inflorescences (I and II) and the chemodiversity α indices of stage IV was also identified. This relationship remained consistent throughout the different stages of the analyzed inflorescences, suggesting that, as the volatile mixture becomes richer and more diversified, it becomes concurrently and proportionally more oxidized (Figure 7).

Furthermore, the analyses indicated a positive correlation between an increase in the Shannon’s chemodiversity of the leaves (richness) and an increase in the frequency of visits by potential pollinators to flowering inflorescences at anthesis (stage III). Additionally, these results reveal a positive correlation between the number of visits by these harmonious insects and the total number of inflorescences.

Another relevant correlation found in the EO of the early stages of the reproductive organ (stages I and II) was found between the microclimatic variable temperature and the chemodiversity α indices. According to these results, the increased temperature resulted in greater richness, abundance, and homogeneity of the EO of these organs.

The analyses also showed a negative correlation between the chemodiversity α indices of the leaves and the microclimatic variable wind. Continuing to regard negative correlations, an inversely proportional relationship was identified between the Simpson’s chemodiversity index (diversity) in the early stages of inflorescence maturation and the total visits of all insects, especially *T. angustula*.

Finally, a negative correlation was also observed between the frequency of visits from some potential pollinators, such as *Meliponini* spp., one species of Syrphidae, and others of Halictidae, and the R&M in stage IV. This suggests that the more oxidized the mixture of volatile compounds at that stage, the lower the frequency of insect visits. Similarly, when the metabolite mixture of the EO is reduced further, there is an increase in the visitation of potential pollinators to the flowering inflorescences. Additionally, the analyses revealed negative correlations between the chemodiversity α indices (leaves and stage II).

## 3. Discussion

### 3.1. Chemical Composition of P. mollicomum Essential Oils

To understand the chemical mediation between *P. mollicomum* and its natural enemies in the Tijuca Forest (Rio de Janeiro), we investigated the chemical composition of EO from leaves and the developmental stages of the reproductive organs of this plant and the correlation between these aspects with visitors and microclimatic environmental conditions. During the early flowering stage, when most specimens studied had immature inflorescences, the predominance of linalool and limonene biosynthesis in these reproductive structures was observed. In this period, the plants were in a phase prior to the proper production of pollen grains, which likely results in pollination limitation. Therefore, we hypothesize that chemical defense of these immature organs is crucial to maintain the reproductive viability of their specimens. Our results are corroborated by previous studies which have suggested that plant species emit toxic volatiles more intensely during the early stages of flowering to protect their pubescent organs against herbivory [39,40]. Additionally, research has shown that genes contributing to the protection of reproductive organs are primarily stimulated during the early stages of the development of young tissues [41]. Previous studies have confirmed limonene’s ability to repel pest organisms, such as aphids, or to attract ladybugs, forming a tritrophic interaction between plant and herbivore and carnivore [42,43,44]. Examining the importance of the constituent linalool, many studies have also highlighted its significant role in attracting insect pollinators [11,24,45,46,47,48]. Additionally, this monoterpene may also have defensive effects against herbivores [49,50,51,52].

The analyses of the final stages of fruiting revealed a greater amount of α-terpineol (11.41%), 1,8-cineole (8.00% to 58.77%), α-pinene (11.30%), β-pinene (6.49%), linalool (11.28%), β-elemene (8.62%), and eupatoriochromene (32.92%). In this stage of development, mature inflorescences stood out, suggesting that the biosynthesis of these constituents is ecologically important, either directly or indirectly, to facilitate reproduction. Previous studies have demonstrated that some of these compounds can play prominent ecological roles, such as in the development of reproductive structures and in protection against herbivores [53,54,55,56]. It is also important to note that the biosynthesis of sesquiterpenes, such as *E*-nerolidol, β-elemene, and germacrene D, was recorded in senescent leaves of *P. mollicomum*, suggesting a protective role for these plant parts [27]. The presence of these constituents in the EO of reproductive organs may also be related to the need to attract the visitation of pollinators and seed dispersers [11,57,58].

For example, the presence of 1,8-cineole in specimens of *P. mollicomum* from the Tijuca Forest has been found for several years [11,30]. Studies have revealed that this monoterpene may possess toxic effects and alter the intestinal microbiota of herbivores and frugivores, thus compromising the digestibility and nutrient absorption of these consumers [59,60,61]. These and other analyses suggest that this major constituent may play an important ecological role in defense against herbivores, increasing in the early stages of inflorescence development and persisting throughout the flowering period [11,30]. Our findings suggest that these volatiles, despite having distinct ecological roles, at the beginning and end of flowering period in *P. mollicomum*, may offer similar functional expression in the chemical communication of this species, perhaps acting synergistically when biosynthesized concomitantly. These hypotheses need to be specifically investigated.

The innovative aspect of this research is its consideration of the way that the ecological roles of these compounds, predominant during the reproductive period in *P. mollicomum*, have often been studied in isolation, disregarding the relevance of minor compounds [11,14,26,30,37]. To improve this discussion, we can draw an analogy between volatiles and soccer players. One star player may have a pivotal role in their team. Then, just as a soccer club typically invests its financial efforts to acquire a valued player, plants also allocate their resources to biosynthesize major and ecologically important substances. However, this player cannot act alone; they depend on other members to form an efficient team. Similarly, “supporting compounds” can have equally crucial importance, acting synergistically with the major compounds and contributing to the survival, acclimatization, and adaptation of plants in their niches; especially when considering that resources, being so scarce, cannot be spent on synthesizing compounds without any function for these plants [38,62,63,64].

### 3.2. Chemodiversity Analysis of P. mollicomum: Ecophysiological and Ecological Interconnections of Volatile Metabolites along Developmental Stages

Small variations in volatile chemodiversity occur over time and space, mainly because EO are directly influenced by biotic and abiotic environmental conditions [38,62,63,64]. For instance, infections caused by insects on vegetative structures can induce the production of new volatile constituents, both locally and systemically; moreover, neighboring plants under herbivore attack may release volatile emissions triggering changes in the EO of adjacent plants [38,62,63,64]. Therefore, studying the volatile chemodiversity and its role as a protagonist in ecological activities is of utmost importance for a comprehensive understanding of its ecological expression.

During the 1980s, O. R. Gottlieb initiated his studies on the evolution of chemical diversity and the oxidation state of compounds in plant species [65,66]. Studies conducted by his group revealed that such phytochemicals play an essential role in the adaptive capacity of organisms to adverse environmental conditions, ranging from high UV radiation to oxygen-rich atmospheres, all factors that may have been crucial for the adaptation of early plants to the terrestrial environment [16,65,66]. From these initial functional traits, the richness of chemodiversity emerged as an intricate phenotypic expression of intra- and interspecific interactions, which are under genetic and environmental control [10,11,16]. These associations among plants of the same and different species promote the biosynthesis of essential metabolites [11,14,16]. Numerous studies have corroborated the importance of these biotic and abiotic factors, highlighting the occurrence of specific chemical variability in response to abrupt environmental changes, such as light radiation [67], temperature [68], herbivory and pollination [37,69], allelopathic influence [70], latitude and longitude [71]; and precipitation [11,30].

Some results of our study reveal the important relationship between temperature, the chemical profile of the reproductive organs, and the chemodiversity of leaves. The increase in temperature was a determining factor for the increase in richness, abundance, and homogeneity of the EO mixture of leaves and reproductive organs. This suggests that thermal conditions may play a crucial role in the biosynthesis and accumulation of volatiles in these plant parts. Our findings corroborate previous studies, which provided specific evidence of how temperature variations can influence the concentration of some terpenes, demonstrating the degradation and expressive biosynthesis of some constituents due to photo-oxidation at high temperatures [72,73]. On the other hand, the results also indicate a negative correlation between the chemodiversity α indices of leaves and the wind microclimatic variable. This relationship suggests that an increase in wind intensity may result in greater volatilization of the chemicals, leading to a reduction in the richness of the EO. 

It is known that some enzymes responsible for terpene biosynthesis are influenced by light, such as for 1,8-cineole synthase and linalool synthase. An in vitro study on volatile constituents at different developmental stages of *Vitis vinifera* L. (Vitaceae) revealed that increased UVB radiation results in a significant increase in the proportion of oxygenated cyclic monoterpenes, such as 1,8-cineole [67]. This observation may also explain our results regarding the decrease in chemodiversity related to the increased volatilization of some constituents, also previously observed [24].

These findings contribute to understanding the underlying mechanisms of production and variation of volatile constituents in plants, considering the influence of abiotic factors. They may have significant implications in the chemical ecology of these plants, as well as in practical applications, such as agricultural management, the production of bioactive substances, and ecosystem studies. However, further research is needed to deepen our understanding of the genetic and/or epigenetic influence on the molecular mechanisms involved in these ecological responses, as well as the potential impact of these changes on plant–environment interaction [62].

This distinction in EO chemodiversity suggests the presence of unique biosynthetic mechanisms regulated by “promiscuous enzymes” in metabolic pathways of each plant organ [4,74,75,76]. As a result, a relevant variety of odoriferous substances is biosynthesized. This complex system is notoriously understood to be controlled by widely studied enzyme superfamilies called terpene synthases (TPS) [4,74,75,76]. Through their activity, an intricate network of mono- and sesquiterpenes can be formed, generating a universe of aromas that play critical roles in plant communication. TPS are even distinguished at phylogenetic levels, being a major theme in chemotaxonomy [4,37,74,75,76,77], and are subjected to years of pressures, selective bottlenecks, and adaptations and mediated by various biotic and abiotic factors [4,37,74,75,76,77].

In the context of plant–pollinator ecology, our results provide valuable insights into the importance of chemodiversity in the specificity of ecological activities. This phenomenon may act by attracting pollinators to locate, recognize, and determine the quality of the reward offered by the plant [4,11,24,27,30,74,75,76]. These findings corroborate the data analyzed in this research, which shows strong positive correlations between the chemodiversity α indices of different plant parts, presenting a fine association between the frequency of visits by potential pollinators and responses to environmental variables. For example, our investigations found that leaves and inflorescences in anthesis (stage III) exhibited the highest chemodiversity indices. These analyses may suggest that the plant possibly directs its allocation of chemical resources with the purpose of defending its still immature organs against herbivores, as well as to attract and reward its mutualistic agents [11,69]. 

Conversely, infructescences showed low chemodiversity indices, possibly indicating a shift in resource redirection towards the biosynthesis of carbohydrates to attract seed dispersers [36,78,79,80,81,82]. These results suggest the possibility of a drain effect between vegetative and reproductive organs [83], where resource allocation may influence the chemodiversity of different plant organs, consequently characterizing each organ as a specific functional trait [11,69].

Furthermore, an increase in chemodiversity by Shannon index, directly related to the increase in the frequency of visits by potential pollinators to inflorescences in anthesis (stage III), was found in our research. It is also important to note that the highest frequency of visits was recorded in the months of October and November, a period during which a greater number of blooming inflorescences (stage III) were quantitatively observed [11]. This finding may suggest the hypothesis that the richness of volatiles present in the EO may positively influence the interaction between plants and their mutualistic agents [11,30]. Also observed, concurrently, was an inversely proportional relationship between the chemodiversity by Simpson index in the early stages of inflorescence maturation and the total visits of all insects, especially *T. angustula* (jataí). The Simpson index represents the diversity of one or more volatile constituents in samples of these stages [12,16,26,38,84]. This latter analysis suggests that the diversity of substances, typically found in the EO of *P. mollicomum* may not be sufficient for insect attraction [11,30,69].

Our studies infer that EO synthesized with higher richness (Shannon index) and evenness (Pielou index) in *P. mollicomum* may be more attractive to insects. Leaves were the most “Equidiverse” * organs (* a term proposed by the authors to express the idea of an abundant and uniform chemical profile). Stages I, II, and III also exhibited high equidiversity, indicating that, at different stages of reproductive organ maturation, they may present a volatile composition with high diversity and uniformity. This is so for two vital reasons: protection of these developing structures (stages I and II), and attraction of insects for the dispersal of their generative whorls (stage III) [11,26,30]. On the other hand, stage IV showed low equidiversity, which allows us to infer that there is a channeling of adaptive metabolism towards survival in these structures, which corroborates the perspectives of the drain effect [83]. For example, a recent investigation, which focused on the analysis of the chemical composition of EO from *Satureja hortensis* L., revealed the presence of a notable diversity and uniformity of volatile constituents in this plant. The research emphasized the synergistic effects demonstrated by the substances, which showed relevant insecticidal properties in combating pests [85].

We can infer that this “phytochemical socialism” is probably crucial, both to attract and/or to repel chemically mediated interactions. The “volatile fog” [11] present in the leaves and in different stages of the reproductive organs, may function as an innate chemical barrier or attractant. If an herbivore manages to overcome the first protective barrier on the leaves, it can find other volatiles, rich and evenly distributed in “chemical weapons”, which act together to minimize the herbivory [11,85,86]. However, as mentioned above, reproductive stage IV showed low equidiversity, indicating that the richness of secondary metabolites may not be a fundamental factor for attracting seed dispersers [36,78,79,81,82]. These results are consistent with previous studies that have investigated the allocation of substances at different stages of plant development, as well as the importance of carbohydrate biosynthesis in attracting dispersers [36,78,79,80,81,82]. In conclusion, the analyses conducted have provided important insights into chemodiversity in different structures and stages of development of *P. mollicomum*.

### 3.3. Similarity between the Chemical Profiles of the Reproductive Organs of P. mollicomum Using the Jaccard Index 

The exploration of the distinct facets of chemodiversity, the focus of our study, has allowed a deeper investigation into the potential dichotomy in the chemical profile of the samples, with emphasis on the different stages of the reproductive organs of *P. mollicomum*. The assessment of chemodiversity β, by Jaccard index, was employed to examine the chemical congruence among the different developmental stages [16,38]. Our results reveal interesting nuances in this context. A similarity of 53% was found between stages I and II, and 50% between stages I and III, as well as 36% between stages I and IV. Other results show stages II and III (55%), II and IV (41%) and, III and IV (55%). These findings indicate that the chemical profile affinity among the different stages is considerably reduced, highlighting the chemical uniqueness of each germinative structure, even if they retain anatomical similarities. This discovery reinforces the notion that different developmental stages of the reproductive organ of *P. mollicomum* possess unique chemical profiles, which may be related to specific physiological and systemic functions at each maturation stage. 

At an individual level, the chemodiversity of a specimen depends on the amount of information present in its vicinity. It is known that the less interference from other biotic interactions, the lower the chemical diversity [12,30,87]. Previous studies conducted by the group have shown that chemodiversity depletes in plant populations removed from their niche [88]. Therefore, the indices quantified in this work play a fundamental role in inferring the characterization of chemical variations, mainly concerning their geographical (different niches), spatial (distinct anatomical structures), and temporal (circadian rhythm) distribution [11,16,27,30].

Thus, understanding and assessing different levels of chemodiversity is essential for comprehending the complex interactions between organisms in their niches. The proper application of these indices provides a valuable quantitative insight into chemical diversity and its role in species ecology and acclimatization, offering a solid subsidy for future ecological and biotechnological studies. When addressing the intricate networks of interactions that permeate the production of volatile constituents, it is essential to consider the potential impact of biotic and microclimatic factors in this process. These elements play a crucial role in the biosynthesis, accumulation, and release of essential metabolites, consequently influencing the attractive response of pollinators or the repellent response of disharmonic insects [11,30]. Understanding these complex “chemo-signaling networks” is of utmost importance for a holistic view of the ecology of ecological activities provided by *P. mollicomum* in its niche. Furthermore, it is well established that these interactions can influence the dynamics of oxidative-reductive patterns present in the EO of these plants [14,16,26].

### 3.4. Ramos and Moreira Index: A Redox Evaluation

It has been recorded in this study that the richer and more diversified the mixture of volatile components, the higher the R&M. The mixture becomes more oxidized when it is chemically more diverse. This was observed in the EO of both leaves and reproductive organ stages, except for stage IV. Dr. Gottlieb’s hypothesis, mentioned above in this text, presumes that plant secondary metabolites function to capture free radicals [16,65,66,89]. The intense enzymatic activity (peroxidases, terpene synthases) in leaves and reproductive stages, especially stage III (anthesis), aims to generate richer and more diversified volatile mixtures, which would also lead to increased production of these free radicals. Consequently, these free radicals (mainly reactive oxygen species (ROS)) are eliminated by mixtures of compounds that include EO, resulting in increased oxidation of the mixture, then reflecting on R&M [16,65,66,89].

## 4. Conclusions

In summary, analyses of the chemical composition of *P. mollicomum* in the neotropical Tijuca Forest revealed important insights into the interactions among volatile constituents, environmental factors, and their potential pollinators. The results indicate that variations in the chemodiversity of the volatile chemical profiles of leaves and different stages of the reproductive organ showed correlations between microclimatic variables and activities of potential pollinator visits, along with significant differences in chemical evenness among these different vegetative structures. These findings reveal the importance of understanding the complex interactions among plant metabolism, its environment, and ecological relationships in its niche, and provides a comprehensive view of the ecology and evolution of ecological activities provided by this specie. We prove that, by inserting chemodiversity indices in the analysis, these interactions can be quantified for better understanding.

## 5. Materials and Methods

### 5.1. Study Area

The area chosen for this research was the Tijuca National Park (TNP) (43°14′29.64″ W, 22°58′9.80″ S), located in the South Zone of Rio de Janeiro city. According to the Köppen–Geiger classification, the TNP has a tropical monsoon climate [90]. Six adult specimens of *P. mollicomum* Kunth, with an average height of 1.65 m, were selected in an open area with elevations ranging from 68 to 127 m. The license permission for conducting these investigations was obtained from the Biodiversity Authorization and Information System (SISBIO—number 57296-1; authentication code 47749568). The experimental plot was surrounded by native vegetation. The fertile specimens were previously identified by the taxonomist Dr. George Azevedo de Queiroz at the Rio de Janeiro Botanical Garden Research Institute (JBRJ), and the samples were deposited in the Herbarium of the State University of Rio de Janeiro—Maracanã Campus (HRJ/UERJ) (Appendix A).

### 5.2. Collection of Microclimatic Data

The averages of the data regarding microclimatic variables were collected weekly and recorded by manual measuring instruments [11]. A digital windmeter anemometer (SIN2919025384—Brazil) was used to measure wind speed, temperature, and relative humidity. Additionally, a luxmeter (INSTRUTEMP, 1712268—Brazil) was employed to measure local luminosity. An infrared thermometer (EXBOM—TDI 330) was utilized to record the surface temperatures of leaves and inflorescences during anthesis. These measurements were taken once at each interval corresponding to the frequency of visits by potential pollinators. Climatological data related to rainfall indices were obtained from the Meteorological Station of Forte de Copacabana, under the jurisdiction of the State of Rio de Janeiro, during the corresponding weeks of observation periods. This institution is affiliated with the Brazilian Meteorological Institute (INMET), responsible for managing climatic information. The analyzed meteorological variables included air temperature (°C), inflorescence temperature (°C), leaf temperature (°C), relative air humidity (%), wind speed (m/s), rainfall (mm), and light intensity (kJ/m^2^).

### 5.3. Evaluation of Pollinators Visit Frequency 

The observations regarding insect visit frequency were conducted weekly from September 2020 to January 2021, from 8:00 a.m. to 5:00 p.m. (with 30 min of surveillance and 30 min of break), totaling 128 h. This period was chosen due to the presence of inflorescences in anthesis (stage III) on the *P. mollicomum* individuals. Frequency counting was performed whenever insects were foraging on the inflorescences during this period [91,92]. The collection of these ecosystemic agents was carried out using an entomological net with a reach of 3 m (basket with approximately 35 cm in diameter, 80 cm in depth, and 3 mm mesh). After capture, the animals were promptly anesthetized in vials containing cotton slightly dampened with 70% (*v*/*v*) hydroalcoholic solution for subsequent identification [93]. Insect descriptions were made using identification keys and entomological specimens from the Reproductive Biology and Pollination Laboratory of the Research Institute of the Botanical Garden of Rio de Janeiro (JBRJ).

### 5.4. Reproductive and Vegetative Phenological Study

Quantitative studies on the reproductive phenology of *P. mollicomum* were conducted throughout the observation of the phenological events: quantities of immature and mature inflorescences, and the total quantity of reproductive organs. Due to the small size of the flowers and fruits, it was necessary to use a manual lens with a magnification of 60× to determine which flowers were in anthesis [94]. The intensity of Fournier (IF) methodology [95] was employed for the quantification of phenophases, which involves creating a scale that classifies the analyzed phenological patterns in plants, previously described in [11].

### 5.5. Essential Oil Obtention and Analysis

The collection of the EO from the leaves (150 g) and the four stages of the reproductive organ (40 g) occurred monthly, between 9:00 a.m. and 10:00 a.m. The samples were comminuted and placed in a 2 L round-bottom flask containing 700 mL of distilled water and subjected to the hydrodistillation method using a modified Clevenger-type apparatus. The process lasted for 2 h. Upon completion, the pure EO was separated from the aqueous phase, dried with anhydrous sodium sulfate, and placed in amber vials for storage in a freezer at −20 °C. The four stages were described by taxonomists Dr. Elsie Franklin Guimarães and Dr. George Azevedo de Queiroz, and they are illustrated in Appendix A and previously characterized [11].

The obtained EO was solubilized in spectroscopic grade dichloromethane from Tedia (Brazil), to reach a final concentration of approximately 1000 μg/mL. These diluted solutions were then subjected to gas chromatography–mass spectrometry (GC–MS) analysis using an HP Agilent GC 6890—MS 5973N instrument to identify volatile constituents by their respective mass spectra. In turn, for the determination of relative percentage parameters and calculation of retention index (RI), the analyses were conducted by gas chromatography–flame ionization detector (GC–FID). The conditions employed in the GC–MS analyses involved the use of an analytical capillary column, HP-5MS (30 m × 0.25 mm i.d. × 0.25 μm, film thickness), with a temperature ramp from 60 °C to 240 °C, increment of 3 °C/min, and helium as carrier gas (~99.99%) at a constant flow rate of 1.0 mL/min. Additionally, the mass scan range (*m*/*z*) ranged from 40 to 600 atomic mass units (u), with an electron impact energy of 70 (eV), operating in positive mode. A 1 μL sample of the EO solution was injected splitless, with the injector temperature set at 270 °C. GC–FID analyses were conducted under the same conditions, though hydrogen (mobile phase with a flow rate of 1 mL/min) and synthetic air were used to produce the flame. Retention times (tR) were measured in minutes, without correction, and the relative percentage of each identified substance was determined based on the signal area. RIs were calculated from the results of the analysis of a homologous series of saturated aliphatic hydrocarbons (C_8_–C_28_), from Sigma-Aldrich, Rio de Janeiro, Brazil, using the same column and conditions employed in the GC–FID analyses. The identification of constituents was performed by comparing the calculated RIs and obtained mass spectra with information available in the literature [96]. GC–FID analyses were conducted in triplicate [11]. Constituents were identified by comparison of their calculated LRIs with those in the literature, and by comparison of the mass spectrum with those recorded by the National Institute of Standards and Technology (NIST) library “NIST14” and Wiley (ChemStation data system) “WILEY7n.” [14]. Additionally, authentic pattern co-injection was performed whenever possible [14].

### 5.6. Chemodiversity Indices Calculation

To quantify the dynamics of the chemical profile of the EO from the leaves and different stages of the reproductive organ, the α and β chemodiversity indices were calculated [12,16,26,38]. It is relevant to mention that these indices were initially developed to assess species diversity on a spatial scale [26,84,97,98,99,100,101,102,103,104,105] and currently have been adapted to measure chemical phenotypic variability [12,16,26,38].

To evaluate chemodiversity α, the following indices were used: Shannon [99], Simpson [84], and Pielou [100]. For chemodiversity β, the Jaccard index was applied to assess the similarity of samples from different stages of the reproductive organ [97]. These levels are classified based on intra- and interspecific chemodiversity categories, as well as richness, abundance, and presence/absence of chemical constituents, and adapted for chemical diversity analyses in different reproductive organs. These quantifications are described below:

Chemodiversity α: This index was adapted to infer the chemical diversity of an individual. The diversity of chemical profiles can be evaluated at different temporal scales or even in a specific tissue [16,38]. This variability can characterize an adaptation or acclimatization of the individual or organ to ecological complexity influenced by positive or negative biotic and abiotic interactions [12,16,38].

The equations for chemodiversity α indices are presented below:

Shannon Index = Σ(Pi × ln (Pi))
(1)


Simpson Diversity Index: = Σ(Pi × ln (Pi − 1))
(2)


Pielou Index: = −Σ(Pi × ln (Pi/ln(S))
(3)

Pi represents the proportional abundance of identified substances, obtained by dividing the relative abundance (or yield of the identified compound) by the total number of determined metabolites, as well as the number of substances in the sample. S is equal to the total number of substances. The natural logarithm (ln) is the logarithm to the base “e” (Euler’s number, approximately 2.71828) × n.

Chemodiversity β: This index was adapted to infer the chemical diversity of different individuals of the same species in a geographically segregated population. Moore and colleagues (2014), from a contemplative perspective, presented the idea that intraspecific chemical changes occur at different levels of diversity, synthesized by distinct metabolic pathways influenced by many variables [38]. Therefore, in this research, this index was developed to evaluate the similarity of chemical profiles among different stages of the reproductive organ of *P. mollicomum*.

To assess chemodiversity β, the Jaccard index was used [4], as described below:

Jaccard Index: (c/(a + b − c))
(4)


In this equation, “a” represents the number of substances in one sample, “b” the number of substances in another sample, and “c” the number of common substances in both samples.

### 5.7. Ramos and Moreira Index (R&M)

In this research, the Ramos and Moreira index (R&M), a chemophenetic index, was also applied. This index offers a quantitative approach to evaluate oxidation-reduction (redox) patterns in mixtures, allowing for a deeper understanding of redox mechanisms in a context that can be applied to chemical ecology studies [16,26].

For the calculations, redox indices were initially used to analyze substances in relation to their oxidation number (NOX), following rules established by Hendrickson–Cram–Hammond [16,26,106], to determine the sum of the oxidation states of each atom in the molecule [16,26,107].

To calculate the weighted mean oxidation-reduction pattern (SRO), the oxidation number (NOX) of the constituent of interest was multiplied by the relative percentage content obtained in the sample analysis (Q%), and this value was subsequently divided by the number of carbon atoms in the molecular skeleton (n) [16,26]. The equation is described below (5):

SRO = (NOX substance × Q%)/n
(5)


This equation provides a weighted average value of the oxidation of the carbon atoms of the mixture constituents (in this case, EO), and represents an intermediate step to obtain the Ramos and Moreira Index (also called General Mixture Redox Index (GMRO)), as defined by Equation (6) [16,26]. To calculate the GMRO, it is necessary to sum the SRO values of all constituents of the mixture and divide by the number of substances identified in the sample (NCI) [16,26]:

R&M = (Σ SRO)/NCI)
(6)


### 5.8. Statistical Analysis

Statistical analyses were conducted using the statistical software Jamovi 2.2.5. To explore the data descriptively, measures of central tendency and dispersion were employed, providing a more precise visualization of the results.

For the final analysis, we developed an innovative graph representing the relationship between richness and evenness. For this, the data obtained from the Shannon index (x-axis) and Pielou index (y-axis) were used, representing respectively the richness and uniformity of the chemical profile of the samples. Based on these data, four distinct ranges were established for categorizing the values “very high” for values between 80% and 90%, “high” for values between 80% and 60%, “medium” for values between 60% and 50%, and “low” for values below 50%. These ranges were referenced from the literature of prominent ecologists such as Robert E. Ricklefs and Relyea Rick in “The Economy of Nature”, and Peroni and Hernandez in “Ecology of Populations and Communities” [108,109]. 

Additionally, the Tukey post-hoc test was performed to obtain a better interpretation of the differences between the means of the compared groups. This test allows for multiple comparisons between samples, identifying which are statistically different from each other. The interpretation of the Tukey test is based on the significance values (*p*-value) obtained for each pairwise comparison. If the significance value is less than a certain pre-established level (usually 0.05), this indicates that there is a statistically significant difference between the compared groups [110]. 

Pearson correlation tests were also conducted to assess relationships between abiotic factors, volatile constituents present in different stages of reproductive organs, and frequency of visitation by potential pollinators. The correlation coefficient of this inspection can be positive (directly proportional) or negative (inversely proportional), assimilating quantitative values that suggest whether the relationship between variables is strong, moderate, or weak, as follows: 0.19 (very weak), 0.20 to 0.39 (weak), 0.40 to 0.69 (moderate), 0.70 to 0.89 (strong), and 0.90 to 1.00 (very strong)

These analyses were conducted in the R program and designed to evaluate the multivariate relationship between the different data, across different collection periods [111].

## Figures and Tables

**Figure 1 plants-13-02497-f001:**
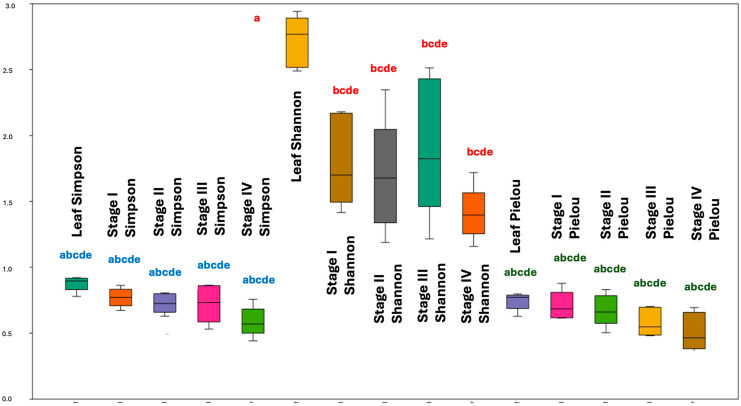
Comparison between means (analysis of variance) of the chemodiversity α indices (Shannon, Simpson and Pielou) between leaves and the different stages of the reproductive organ of *P. mollicomum*. Leg. Same letter indicates no statistical difference (*p* > 0.05).

**Figure 2 plants-13-02497-f002:**
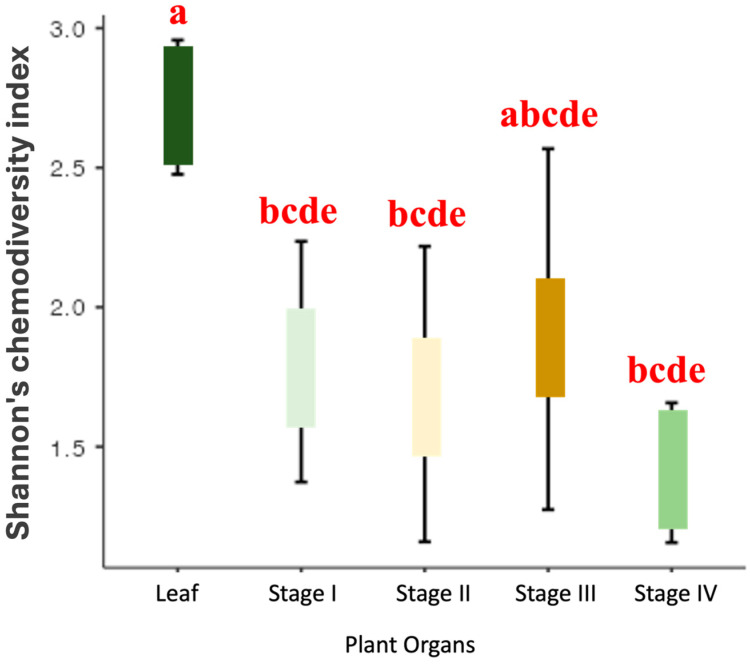
Tukey test (ANOVA) of the means of Shannon’s chemodiversity index for leaves and reproductive organ stages of *P mollicomum*. Leg. Same letter indicates no statistical difference (*p* > 0.05).

**Figure 3 plants-13-02497-f003:**
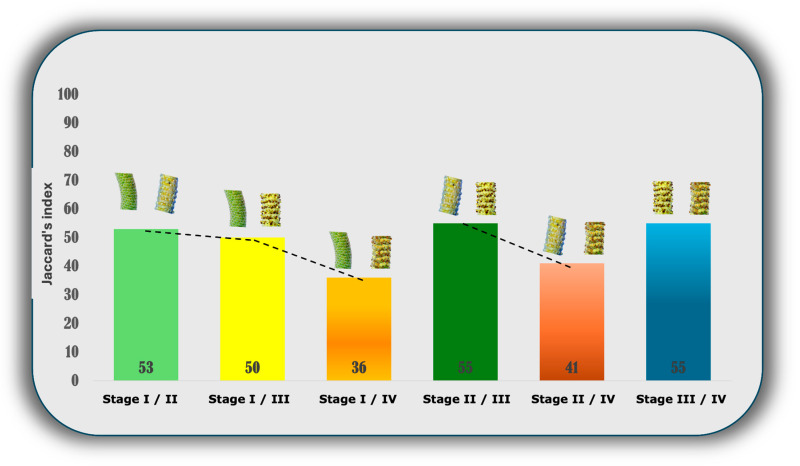
Jaccard index for assessing chemodiversity β among the distinct stages of the reproductive organ of *P. mollicomum*. Leg. numbers in the bars indicate similarity percentual.

**Figure 4 plants-13-02497-f004:**
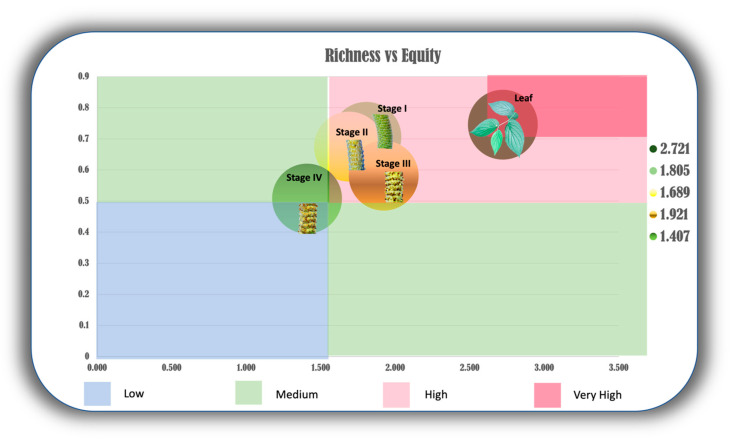
Scatter plot representing two chemodiversity α indices: Shannon (richness) and Pielou (evenness). Leg. The y-axis represents the Shannon index and the x-axis represents the Pielou index.

**Figure 5 plants-13-02497-f005:**
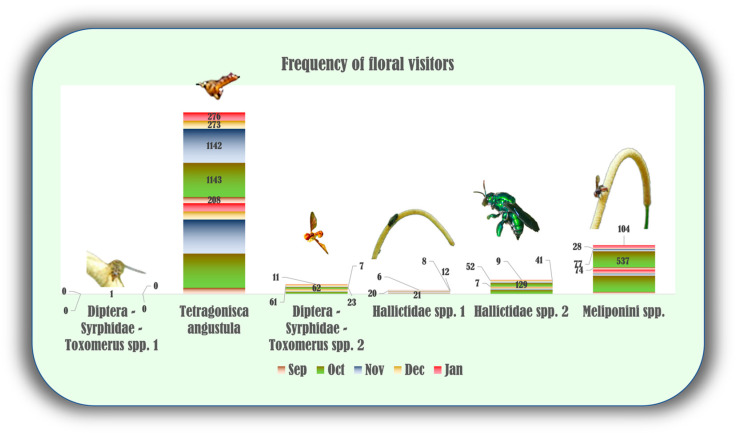
Visit frequencies of potential pollinators of *P. mollicomum* from Tijuca Forest/RJ (September 2020 to January 2021). Leg. The x-axis represents the visit frequencies of potential pollinators, and the y-axis represents the months under analysis. Sep—September, Oct—October, Nov—November, Dec—December, Jan—January.

**Figure 6 plants-13-02497-f006:**
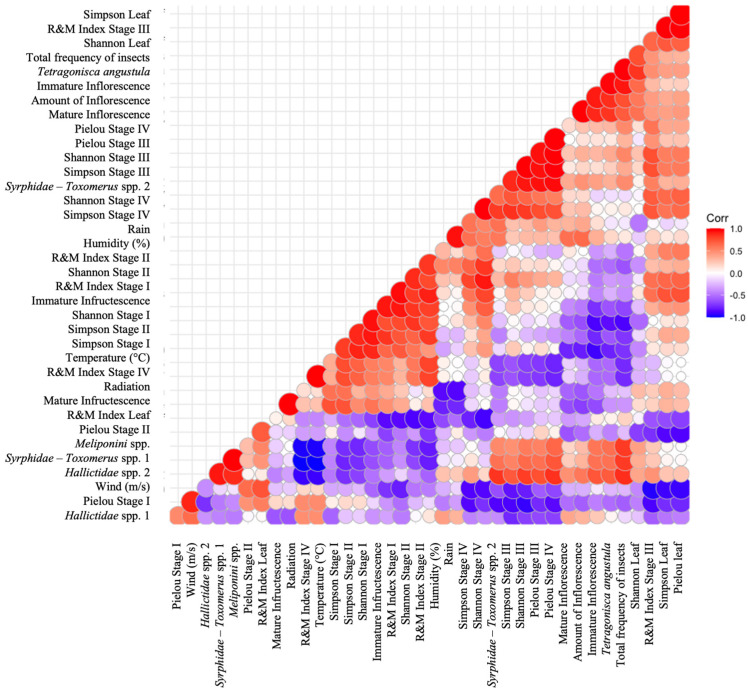
Pearson correlation analyses regarding the following variables: insect visit frequency, chemodiversity α indices, Ramos and Moreira redox indices, phenology (quantity of reproductive organs), and microclimatic factors. Leg. Corr—Pearson correlation; R&M—Ramos and Moreira index.

**Figure 7 plants-13-02497-f007:**
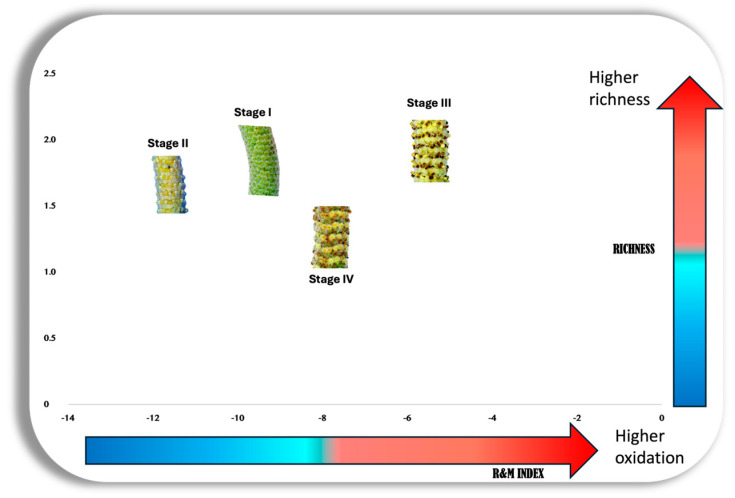
Scatter plot depicting the relationship between the Shannon index (Chemical richness) and the Ramos and Moreira index (R&M, redox pattern) of the essential oils from different stages of the reproductive organ of *P. mollicomum*.

## Data Availability

Data is contained within the article and Appendix A.

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
