# Peer review of "From Leaves to Reproductive Organs: Chemodiversity and Chemophenetics of Essential Oils as Important Tools to Evaluate Piper mollicomum Kunth Chemical Ecology Relevance in the Neotropics"

_plants, 2024, doi:10.3390/plants13172497_

Round 1

Reviewer 1 Report

Comments and Suggestions for Authors

see attach

Comments on the Quality of English Language

Minor editing of English language is required.

Author Response

Response to Reviewer 1 Comments

- This is a valuable paper dealing with experiments directed to establish possible relationship of the chemical diversity of the essential oils (EOs) of different organs of Piper mollicomum and climatic variables to the attraction of possible pollinators to the plant. The results of the work confirm the authors expectations and provides a nice advance in the knowledge of interactions among volatile components, environmental agents and potential pollinators in neotropical forests. The ms is worth of publication in Plants, but I feel highly concerned about how the authors have identified the chemicals in tables S1, S2 and S3. Apparently, they have just characterized the chemical components of the oils by GC-FID (retention index, RI, vs the RI of the compound in the literature) and GC-MS (by comparison the mass spectrum of the unknown compound vs that of the literature). In many cases, researchers have to use also authentic synthetic compounds to confidently assign the correct structure to an unknown compound, but in this ms there is no comment about this possibility even when the RI of the unknown compound is quite different to that of the literature. This is the case of many compounds in tables S1, S2 and S3. F.i., myrcene (difference of RIs 11), βelemene (dif. 12), eupatoriochromene (dif. 29), benzyl benzoate (dif. 16), β-pinene (dif. 35), myrcene (dif. 43), linalool (dif. 80!), tridecanone (dif. 24), β-panasinsene (dif. >100!), and many more. To me, most of these assignments are probably wrong, and I suggest the authors to exclude from the tables those compounds with a difference in RI ≥10 or in case of doubt re-check using an authentic sample. This must be done in a revised version of the ms and the authors should be careful with the comments of such compounds in text if their chemical assignment is not certain. - We deeply appreciate your meticulous comments highlighting the importance of accurate identification of chemical constituents. We fully agree that the characterization of essential oil (EO) constituents is crucial to ensuring the reliability of the results presented. In our methodology, we followed a robust protocol that includes the use of internal standards for the authentication of key volatile compounds such as linalool, caryophyllene, and 1,8-cineole. These standards were carefully selected and compared with all analyzed samples, reinforcing the reliability of the assignments made through GC-FID and GC-MS. However, we did not use internal standards for all identified and quantified substances, which is why this was not mentioned in the original text. Nonetheless, we have now included in the methodology that we used internal standards to ensure authenticity.

It is important to note that variations in retention indices (RI)—as observed for compounds like myrcene, β-elemene, and linalool—can occur due to a range of technical factors such as the choice of chromatographic column, injection conditions, applied temperature ramp, and carrier gas flow. These variations are well-documented in the scientific literature and are considered normal within certain limits, as described in the following studies:

  • Babushok, V.I., Zenkevich, I.G. Retention Indices for Most Frequently Reported Essential Oil Compounds in GC. Chroma 69, 257–269 (2009). https://doi.org/10.1365/s10337-008-0872-3
  • Shellie R, Mondello L, Marriott P, Dugo G. Characterisation of lavender essential oils by using gas chromatography-mass spectrometry with correlation of linear retention indices and comparison with comprehensive two-dimensional gas chromatography. J Chromatogr A. 2002 Sep 13;970(1-2):225-34. doi: 10.1016/s0021-9673(02)00653-2. PMID: 12350096.
  • Sadgrove, N.J.; Padilla-González, G.F.; Phumthum, M. Fundamental Chemistry of Essential Oils and Volatile Organic Compounds, Methods of Analysis and Authentication. Plants 2022, 11, 789. https://doi.org/10.3390/plants11060789
  • Tedone L, Bonaccorsi I, Dugo P, Cotroneo A, Dugo G, Mondello L. Reliable Identification and Quantification of Volatile Components of Sage Essential Oil Using Ultra HRGC. Natural Product Communications. 2011;6(3). doi:10.1177/1934578X1100600321

These recent studies demonstrate that the use of different columns (e.g., columns with different stationary phases or lengths) can lead to significant variations in RI without compromising the correct identification of compounds, especially when corroborated by mass spectrometry. For example, the study by Shellie et al. (2002), cited above, showed that different stationary phases and columns can result in significant variations in RI without compromising the accuracy of identification when corroborated by mass spectrometry. Additionally, the study by Babushok and Zenkevich (2009) provides a detailed analysis of retention index variations for compounds commonly found in essential oils, suggesting that such variations are expected within certain technical limits.

Moreover, the NIST database already recognizes this acceptable variation (https://webbook.nist.gov/cgi/cbook.cgi?ID=C470826&Mask=2000#Gas-Chrom). We have included these data with the described variations and acknowledge that it is not uncommon to use RI or MS data obtained from the literature of ADAMS, WILYS, and/or NIST to identify compounds, including variations of 5 to 100 indices from the main literature used, which is ADAMS.

However, we acknowledge the importance of validating these assignments with authentic samples whenever possible. While we used internal standards for major compounds, authentic samples for all mentioned compounds were not available at the time of analysis. We plan to include this step in future studies to further ensure the accuracy of our identifications, as well as conduct structural determination analyses, which are not the focus or objective of this research that has a robust chemical-ecological approach.

We would like to emphasize that the observed variations were discussed in detail in the manuscript, and the identifications were made based on multiple lines of evidence, including mass spectra, calculated RI, and specialized bibliography. We assure you that all precautions were taken to ensure the accuracy of the chemical assignments and that the compounds were identified with the utmost rigor possible within the available technical and material limitations. We were scientifically correct in presenting the data as we obtained them, corroborating with different studies that used conditions similar to this study (see references in Ramos et al., 2020; Costa-Oliveira et al., 2023).

We once again thank you for your comments and are available to make any necessary revisions to further strengthen the quality of the manuscript.

Other minor points that should be considered by the authors in a revised version are the following:

  1. In the Abstract, for identification of the EOs the authors use analyses only by gas chromatography. However, in MM and tables S1-S3 they also cite gas chromatography coupled to mass spectrometry, an essential technique although not sufficient to assess full identification of chemical compounds, as cited above. The authors should also include this technique in the Abstract - I apologize for the oversight. Thank you for your careful review, and I would like to inform you that the error has been corrected in the manuscript. The technique of gas chromatography coupled with mass spectrometry is now appropriately mentioned in the Abstract, as well as the Gas Chromatography-Flame Ionization Detector (GC-FID), which was also previously omitted. – Done.

  1. In table S3 it is included (E)-3-hexenol. The Z isomer is much more common in nature. Please, check by using the commercially available synthetic material of the latter and determine its RI. Why in table S3 ‘calculated retention index’ is tR and ‘literature retention index’ is IRlit while in tables S1 and S2 the same indexes are RIcalc and RIlit?. In table S4 comparison of the Shannon index of leaves and stage II is 0.055. Can this be considered statistically significant (p<0.05)? –
  2. We sincerely appreciate your meticulous observation regarding the compound listed in Table S3. After re-evaluating the literature and our data, we identified that there was indeed a typographical error, where (E)-3-hexenol was mistakenly reported instead of the more common (Z)-3-hexenol. The retention index (RI) provided in the table was correctly associated with (Z)-3-hexenol. This error has been corrected in the revised manuscript. We apologize for any confusion this may have caused.
  3. Thank you for pointing out the inconsistency in the nomenclature of retention indices between Tables S1, S2, and S3. You are correct that the retention time (tR) was mistakenly labeled as 'calculated retention index' in Table S3. The appropriate designation should indeed be 'retention time' (tR). We have corrected the table legend in the supplemental material to ensure consistency across all tables, aligning with the terms 'RIcalc' for calculated retention indices and 'RIlit' for literature retention indices as used in Tables S1 and S2.
  4. We appreciate your query regarding the statistical significance of the Shannon index comparison between leaves and stage II in Table S4. Statistical significance is indeed influenced by the variability of the data and the statistical tests employed. In our study, the analysis conducted with the statistical software, as described in the methodology, determined that the observed difference is statistically significant, with a p-value of <0.05. Although the absolute difference in the Shannon index may appear small (0.055), the statistical test accounts for the data's inherent variability, confirming the result as significant within the study's context. We appreciate your insight and hope this explanation clarifies the statistical relevance of our findings.

  1. L47, delete ‘in and an’ to say ‘present at different levels’. - Done.
  2. L72,74, delete ‘in 2020’ and ‘data from the author and literature’ and just insert the corresponding ref - Done.
  3. L73 and elsewhere, the acronym EO in ‘essential oils (EO)’ should be plural - Done.
  4. 102 and elsewhere, name of plants should be in italics. – Done.
  5. L105, replace ‘Most’ by ‘Along’. - Done.
  6. L107 and elsewhere, replace ‘in the different stages’ by ‘at the different stages’. - Done.
  7. L118, change order of the sentence by saying ‘In January, the last month of the research - Done
  8. L177, to exhibit. - Done.
  9. L228, delete ‘to complement the findings’ (unnecessary). - Done.
  10. L242, Higher oxidation…? (something is missing). – Deleted.
  11. L284, ‘against in inhibit herbivores’?. Please, revise. – Corrected.
  12. L295-7, I do not think it is appropriate here the comment regarding the possible analogy between volatiles and soccer players. Please, delete. - Thank you for your valuable suggestion. We appreciate your attention to detail and understand your perspective. However, we would like to retain the analogy between volatiles and soccer players. This choice is part of our group's scientific and technical writing style, aimed at adding a distinct personality to our text. Additionally, we recognize that scientific writing can often feel overly standardized, and we believe that analogies like this can engage readers. However, if you still feel that it should be removed, we will make the necessary changes later on.
  13. L315, remove ‘Dr.’ – Done.
  14. L328, remove ‘stand out’. - Done.
  15. L482, change order: ‘from 68 to 127 m - Done.
  16. L537, replace ‘ppm’ by, f.i., ‘mg/mL’ or whatever appropriate. - Done.
  17. L578, 580, 582, in equations 1), 2), and 3) the second parenthesis is over. Also, in L598 – Corrected
  18. L663, delete ‘In other words’, it is colloquial and unnecessary. - Done

Title of references 4,12, 13, 16, 28, 36, 63, 66, 68, 72, 73, 74, 75, 76, 87, 97, 98, 99, 100, 103, 105, 107 are in abbreviated form. - Corrected

Reviewer 2 Report

Comments and Suggestions for Authors

This manuscript provides the comprehensive analyses on chemical diversity of Piper mollicomum and its relation to d environmental factors and the activities of potential pollinators. The results are a significant contribution to chemical ecology of plant and pollinators in a relatively large scale, compared to the most y studies of chemical ecolog with smaller scale. Authors are encouraged to address some concerns for improvement of the manuscript:

1) The title of the manuscript is to general. Actually only chemicals of plant essential oils were analyzed in this study, and these chemicals can not represent all secondary metabolite chemicals, epecially volatile chemicals from plants. So I suggest to narrow the chemical range in the title, for example, as "... Chemodiversity and Chemophenetic of Essential Oils s..." 

2) Abstract can be improved by adding some details of results, and first sentence can be removed and methods can be shortened.

3) Introduction: please add a brief introduction to  Brazilian Atlantic Forest and main composition of plant species, prior to description of importance and reasons for choosing  Piper mollicomumas a model.

4) Resolutions of some figures should be improved.

5) Conclusions can be placed following Results.

Comments on the Quality of English Language

This manuscript is generally well written, and mispelling or grammatical errors may occur, so please check the thoroughout manuscript.

Author Response

Response to Reviewer 2 Comments

We greatly appreciate your positive comments and the detailed evaluation of our manuscript. I am very pleased to know that you consider our comprehensive analysis of the chemical diversity of Piper mollicomum and its relationships with environmental factors and potential pollinators a significant contribution to chemical ecology.

We are attentive to your suggestions and are working to address them as outlined below:

  • The title of the manuscript is to general. Actually only chemicals of plant essential oils were analyzed in this study, and these chemicals can not represent all secondary metabolite chemicals, epecially volatile chemicals from plants. So I suggest to narrow the chemical range in the title, for example, as "... Chemodiversity and Chemophenetic of Essential Oils s..." - Done

  • Abstract can be improved by adding some details of results, and first sentence can be removed and methods can be shortened. - Done

  • Introduction: please add a brief introduction to  Brazilian Atlantic Forest and main composition of plant species, prior to description of importance and reasons for choosing  Piper mollicomum as a model. -

  • Resolutions of some figures should be improved. – Done

  • Conclusions can be placed following Results. – Done
